# CapNav: Towards Robust Indoor Navigation with Description-First Maps

## Abstract

Humans naturally form mental maps of their surroundings: they picture what a destination looks like, relate it to nearby objects, and implicitly plan a route before moving. We seek a similar capability for embodied agents: given a free-form description such as "go to the white sofa with curved edges", the agent should pick the correct 3D instance among many lookalikes and navigate to it safely. We propose *CapNav*, a description-first navigation framework that builds an instance-centric 3D map from RGB-D streams and uses natural-language object descriptions as the primary interface for goal selection. CapNav maintains a dense semantic voxel map for global geometry and, in parallel, constructs persistent 3D object tracks by aggregating Detic based open-vocabulary detections and LSeg features over time. For each stabilized track, a small set of views is captioned with a vision-language model and embedded with BGE-M3, yielding a caption-enriched representation that links language, semantics, and 3D pose. At test time, free-form instructions are encoded in the same text space, matched against object captions to select a target instance, and then converted into a metric-space waypoint followed by A* planning. CapNav shows consistent improvements over category-only and map-based baselines (ZSON, LM-Nav, CLIP on Wheels (CoW), VLMaps) in multi-object navigation tasks, and its instance-level captions make retrieval decisions transparent and easy to interpret.

## 1 Introduction

People navigate indoor spaces by imagining what a destination looks like and where it sits relative to other objects: "the white sofa with curved edges near the window" is often more natural than "sofa." We study *description-based language navigation*, where an embodied agent must follow such free-form instructions, disambiguate between multiple similar instances, and reach the correct 3D object. This is harder than standard vision-and-language navigation (VLN) because it demands attribute-level reasoning (color, shape, material, context) and tight coupling between semantics and metric-space planning. Prior language-informed navigation and open-vocabulary mapping methods partly address this but are mostly not instance-centric. LM-Nav Shah et al. (2023) plans over a topological image memory using an LLM, CLIP on Wheels (CoW) Gadre et al. (2022) and CLIP Map Taguchi & Deguchi (2024) rely on CLIP features for open-vocabulary goal specification, and VLMaps Huang et al. (2023) lifts vision–language features into a 3D voxel grid Toumieh & Lambert (2022). These approaches operate on dense feature volumes or map cells, where attributes are only implicit, leading to difficulties in scenes with many lookalike objects and a loose connection between perception and geometric planning. We propose *CapNav*, a description-first, instance-centric navigation framework that makes object instances the main interface between language and space. From RGB–D streams with known poses, CapNav builds a dual-layer representation: (i) a dense semantic voxel map from LSeg features Li et al. (2022) for global geometry, and (ii) persistent 3D object tracks (SG3DTracker) that aggregate Detic-based open-vocabulary detections and instance masks Zhou et al. (2022) over time. Each stabilized track is summarized by multi-view crops, captioned by a vision–language model and embedded with BGE-M3 Chen et al. (2024), yielding caption embeddings tied to 3D poses and category predictions. At test time, free-form instructions are encoded in the same text space, matched to object captions, optionally reranked by a multimodal LLM, and finally converted to metric-space waypoints followed by A* Guruji et al. (2016) planning. In this way, language-expressed attributes directly shape goal selection, while planning

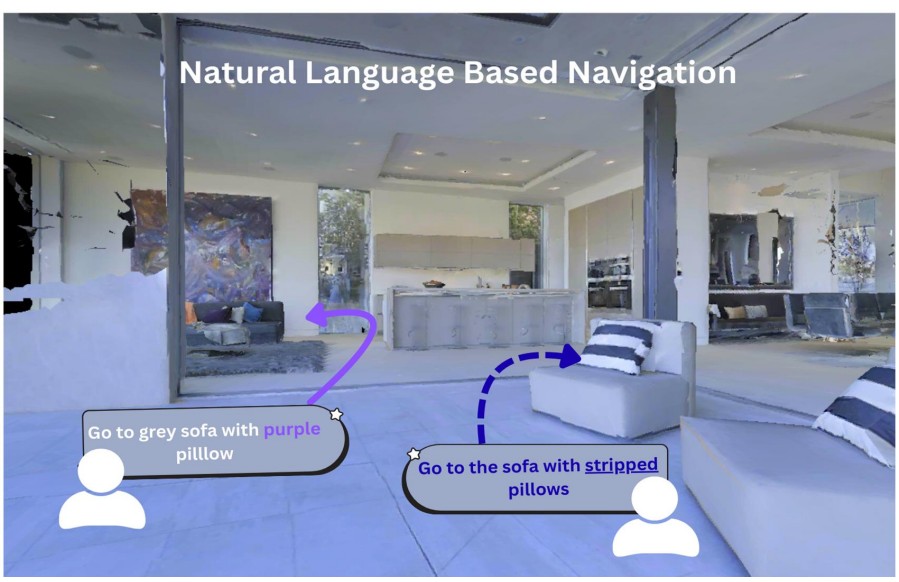

Figure 1: Illustration of natural language attribute-guided navigation.

remains geometric and decisions remain interpretable. Experiments on multi-object navigation in Habitat Savva et al. (2019) with Matterport3D scenes Chang et al. (2017) show that CapNav outperforms category-only and map-based baselines (ZSON, LM-Nav, CLIP on Wheels (CoW), VLMaps) in diverse indoor environments, while its instance-level captions provide clear and interpretable grounding for retrieval-based navigation decisions.

## 2  RELATED WORK

Instruction-guided navigation in 3D environments has been explored through a combination of foundation models, open-vocabulary scene representations, and structured map learning. LM-Nav Shah et al. (2023) uses a large language model to decompose long-horizon instructions into semantic subgoals and grounds them on a topological image memory using vision–language matching, while CLIP on Wheels (CoW) Gadre et al. (2022) employs CLIP-based saliency and exploration to locate open-vocabulary object goals without task-specific training. Beyond purely view-based methods, CLIP Map Taguchi & Deguchi (2024) aggregates CLIP embeddings from egocentric observations into a global spatial representation and retrieves locations based on text–image similarity, and VLMaps Huang et al. (2023) fuses pretrained vision–language features into a 3D voxel grid so that free-form language queries can be localized directly in metric space. These approaches demonstrate that pretrained vision–language models and open-vocabulary features can effectively support language-conditioned navigation, but they typically operate on dense feature volumes or map cells rather than on stable, object-centric entities, making it difficult to reason explicitly about individual instances and their attributes. A complementary line of research focuses on learning structured maps tailored to navigation under instructions. Weakly supervised multi-granularity map learning Chen et al. (2022) enriches semantic maps with finer object-level details and attributes to better localize instruction-relevant regions, BevBert An et al. (2022) pretrains agents on bird's-eye-view maps that combine geometry, appearance, and topological connectivity for long-horizon reasoning, and instruction-guided path planning over 3D semantic maps Wang et al. (2025) formulates navigation as ranking candidate trajectories based on their compatibility with a language description. Collectively, these approaches show that pretrained vision–language models, open-vocabulary features, and structured spatial representations substantially improve language-conditioned navigation, but object attributes such as color, shape, and texture are typically encoded only implicitly within holistic embeddings or coarse map features rather than as explicit, composable constraints tied to persistent 3D instances. The present work focuses on making such attributes explicit at the instance

level in a 3D map and on using multiview-consistent, caption-based attribute descriptors together with language-derived constraints to support more precise goal selection under natural language instructions, especially in scenes with many visually similar objects.

## 3 METHODOLOGY

CapNav builds an instance-centric representation on top of dense semantic lifting so that each object instance is anchored in metric space and directly linked to a language representation. Given synchronized RGB–D streams and camera poses, CapNav constructs: (i) a dense voxel grid storing geometry and per-voxel semantic features from LSeg Li et al. (2022), and (ii) a set of 3D object tracks maintained by a semantic–geometric tracker (SG3DTracker) that aggregates Detic-based open-vocabulary detections and instance masks Zhou et al. (2022) over time. After mapping, each stabilized track is summarized by a small set of multi-view crops, captioned by a vision-language model (Gemini 2.5 Flash Lite Comanici et al. (2025)), and embedded with BGE-M3 Chen et al. (2024), yielding a caption-enriched, instance-level map that can be queried with free-form language and converted to navigation goals. An overview of the pipeline is shown in Figure 2.

### 3.1 PER-FRAME GEOMETRIC AND SEMANTIC LIFTING

At time step $t$, the agent observes an RGB image $I_t \in \mathbb{R}^{H \times W \times 3}$, a depth map $D_t \in \mathbb{R}^{H \times W}$, and a camera pose $T_{c,t}^w \in SE(3)$ mapping camera to world coordinates.

**Geometric back-projection.** We use a standard pinhole camera model

$$K = \begin{bmatrix} f_x & 0 & c_x \\ 0 & f_y & c_y \\ 0 & 0 & 1 \end{bmatrix}. \tag{1}$$

For a pixel $(u, v)$ with valid depth $z = D_t(v, u)$, we back-project into the camera frame

$$\mathbf{x}_t^c(u, v) = \begin{bmatrix} \frac{(u-c_x)z}{f_x} \\ \frac{(v-c_y)z}{f_y} \\ z \\ 1 \end{bmatrix}, \tag{2}$$

and transform to the world frame:

$$\mathbf{x}_t^w(u, v) = T_{c,t}^w \mathbf{x}_t^c(u, v), \qquad \mathbf{x}_t^w(u, v) = [x_w, y_w, z_w, 1]^\top. \tag{3}$$

**Voxel-based semantic fusion.** We discretize the world into a voxel grid $\mathcal{G}$ with resolution $s$. A 3D point $\mathbf{x}^w = [x_w, y_w, z_w, 1]^\top$ maps to voxel indices

$$i = \left\lfloor \frac{x_w - x_0}{s} \right\rfloor, \quad j = \left\lfloor \frac{y_w - y_0}{s} \right\rfloor, \quad k = \left\lfloor \frac{z_w - z_0}{s} \right\rfloor, \tag{4}$$

where $(x_0, y_0, z_0)$ is the grid origin. Each voxel $g$ stores RGB color $\mathbf{c}_g$, a semantic feature vector $\mathbf{f}_g$, and an accumulation weight $w_g$.

We apply an LSeg encoder to $I_t$ to obtain a dense feature map $\Phi_t \in \mathbb{R}^{d \times H' \times W'}$. For each valid pixel $(u, v)$, we sample a feature $\mathbf{f}_t(u, v)$ from $\Phi_t$ and associate it with the corresponding 3D point. A radial weight

$$\alpha(\mathbf{x}_t^c) = \exp\left(-\frac{\|\mathbf{x}_t^c\|_2^2}{2\sigma^2}\right) \tag{5}$$

downweights far or noisy observations. The voxel state is updated by a running weighted average:

$$\mathbf{f}_g^{\text{new}} = \frac{w_g^{\text{old}} \mathbf{f}_g^{\text{old}} + \alpha(\mathbf{x}_t^c) \mathbf{f}_t(u, v)}{w_g^{\text{old}} + \alpha(\mathbf{x}_t^c)}, \tag{6}$$

$$\mathbf{c}_g^{\text{new}} = \frac{w_g^{\text{old}} \mathbf{c}_g^{\text{old}} + \alpha(\mathbf{x}_t^c) \mathbf{c}_t(u, v)}{w_g^{\text{old}} + \alpha(\mathbf{x}_t^c)}, \tag{7}$$

$$w_g^{\text{new}} = w_g^{\text{old}} + \alpha(\mathbf{x}_t^c). \tag{8}$$

As the agent moves, this yields a dense semantic voxel map capturing geometry and coarse appearance cues.

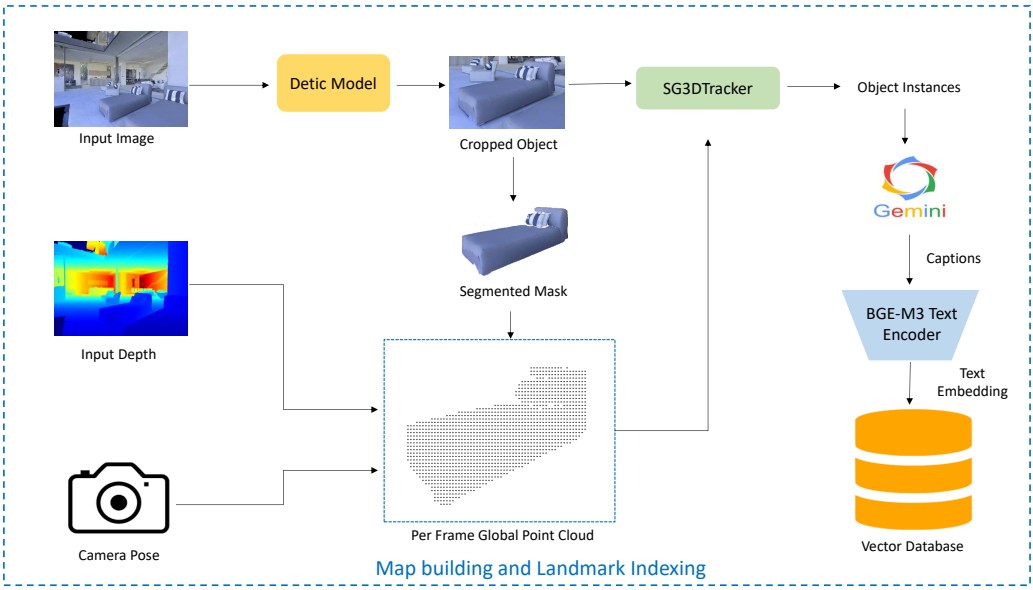

Figure 2: Overview of the CapNav pipeline. RGB–D observations are lifted into a semantic voxel map and tracked as 3D object instances. After mapping, multi-view crops of each instance are captioned and embedded, forming a caption-enriched object database that supports language-conditioned retrieval and navigation.

## 3.2 SEMANTIC–GEOMETRIC 3D OBJECT TRACKING

The voxel grid provides global context, but navigation decisions are made at the object level. CapNav maintains a set of 3D tracks that are updated per frame using geometric and semantic information.

**Detection and lifting.** Given $I_t$, we apply Detic Zhou et al. (2022) in open-vocabulary mode to obtain detections

$$\mathcal{D}_t = \{(b_i, c_i, \ell_i, s_i)\}_{i=1}^{N_t}, \tag{9}$$

with bounding box $b_i$, class ID $c_i$, textual label $\ell_i$, and confidence $s_i$. Detic also predicts an instance mask $M_i$. We compute the mask centroid $(\hat{u}_i, \hat{v}_i)$ and back-project it to $\mathbf{x}_i^w$ using the same intrinsics and pose. An instance-level descriptor is obtained by aggregating LSeg features inside the mask region $\mathcal{R}_i$:

$$\mathbf{s}_i = \frac{1}{|\mathcal{R}_i|} \sum_{(u', v') \in \mathcal{R}_i} \Phi_t(:, u', v'). \tag{10}$$

Each detection is thus $(\mathbf{x}_i^w, b_i, c_i, \mathbf{s}_i)$.

**Data association (SG3DTracker).** We maintain active tracks $\mathcal{T}_t = \{T_j\}$, where each track $T_j$ stores position $\mathbf{p}_j$, velocity $\mathbf{v}_j$, smoothed box $B_j$, mean feature $\bar{\mathbf{s}}_j$, class $c_j$, and lifecycle statistics (age, hits, misses). Between frames, positions are predicted with a constant-velocity model:

$$\mathbf{p}_j^{\text{pred}} = \mathbf{p}_j + \mathbf{v}_j \Delta t. \tag{11}$$

We build a cost matrix $C$ combining 3D distance, 2D overlap, semantic similarity, and class consistency:

$$C_{j,i} = \alpha_{3D} \frac{\|\mathbf{p}_j^{\text{pred}} - \mathbf{x}_i^w\|_2}{d_{\max}} + \beta_{2D}\big(1 - \text{IoU}(B_j, b_i)\big) + \gamma_{\text{sem}}\big(1 - \cos(\bar{\mathbf{s}}_j, \mathbf{s}_i)\big) + \lambda_{\text{cls}} \mathbb{1}[c_j \neq c_i], \tag{12}$$

where IoU is bounding-box IoU and $\cos(\cdot, \cdot)$ is cosine similarity. We fix $\alpha_{3D}{=}0.6$, $\beta_{2D}{=}0.2$, $\gamma_{\text{sem}}{=}0.2$, and $\lambda_{\text{cls}}{=}0.3$ in all experiments. Pairs exceeding $d_{\max}$ or an association threshold $\tau_{\text{assoc}}$

---

**Algorithm 1** SG3DTracker update for a single frame $t$

---

Active tracks $\mathcal{T}_{t-1}$; detections $\mathcal{D}_t = \{(\mathbf{x}_i^w, b_i, c_i, \mathbf{s}_i)\}$. Updated tracks $\mathcal{T}_t$; set of confirmed objects.

**1. Prediction step** *[r]constant-velocity 3D prediction track $j \in \mathcal{T}_{t-1}$ $\mathbf{p}_j^{\text{pred}} \leftarrow \mathbf{p}_j + \mathbf{v}_j \, \Delta t$

**2. Association step** *[r]build and gate cost matrix Build the cost matrix $C_{j,i}$ using Eq. equation 12 pair $(j, i)$ $\|\mathbf{p}_j^{\text{pred}} - \mathbf{x}_i^w\|_2 > d_{\max}$ **or** $C_{j,i} > \tau_{\text{assoc}}$ $C_{j,i} \leftarrow +\infty$ *[r]gate invalid associations Solve the linear assignment on $C$ to obtain matches $\mathcal{M}$ and unmatched sets $\mathcal{U}_T$ (tracks), $\mathcal{U}_D$ (detections)

**3. Matched track update** *[r]update geometry and semantics $(j, i) \in \mathcal{M}$ $\mathbf{p}_j^{\text{prev}} \leftarrow \mathbf{p}_j$ $\mathbf{p}_j \leftarrow \alpha_{\text{pos}} \mathbf{x}_i^w + (1 - \alpha_{\text{pos}}) \mathbf{p}_j^{\text{pred}}$ $\mathbf{v}_j \leftarrow \mathbf{p}_j - \mathbf{p}_j^{\text{prev}}$ $B_j \leftarrow (1 - \alpha_{\text{bbox}}) B_j + \alpha_{\text{bbox}} b_i$ $\bar{\mathbf{s}}_j \leftarrow \text{EMA}(\bar{\mathbf{s}}_j, \mathbf{s}_i)$ $\text{hits}_j \leftarrow \text{hits}_j + 1$ $\text{misses}_j \leftarrow 0$

**4. Track birth (new objects)** $i \in \mathcal{U}_D$ Initialize a new track $j$ from $(\mathbf{x}_i^w, b_i, c_i, \mathbf{s}_i)$ with $\mathbf{p}_j \leftarrow \mathbf{x}_i^w$, $\mathbf{v}_j \leftarrow \mathbf{0}$ $B_j \leftarrow b_i$, $\bar{\mathbf{s}}_j \leftarrow \mathbf{s}_i$ $\text{age}_j \leftarrow 1$, $\text{hits}_j \leftarrow 1$, $\text{misses}_j \leftarrow 0$ Add $j$ to $\mathcal{T}_{t-1}$

**5. Track death (lost objects)** $j \in \mathcal{U}_T$ $\text{age}_j \leftarrow \text{age}_j + 1$ $\text{misses}_j \leftarrow \text{misses}_j + 1$ $\text{misses}_j > m_{\max}$ remove track $j$ from $\mathcal{T}_{t-1}$

**6. Export confirmed tracks** *[r]instances for captioning and navigation Let $\mathcal{T}_t$ be the remaining set of tracks after birth/death updates Return as confirmed objects all tracks $j \in \mathcal{T}_t$ with $\text{hits}_j \geq h_{\min}$ and $\text{misses}_j = 0$, together with their mean 3D positions and voxel-grid indices

---

are gated with $C_{j,i} = +\infty$. Solving the linear assignment problem on $C$ gives matches; unmatched tracks and detections trigger track death and birth. Matched tracks update position, velocity, box, and mean feature via exponential moving averages, and their hit/miss counters are updated. Tracks missed for more than $m_{\max}$ frames are removed; tracks with at least $h_{\min}$ hits and no recent misses are exported as confirmed objects.

Algorithm 1 summarizes one update of SG3DTracker. Its output is a set of stable 3D instances with consistent IDs, poses, category predictions, and descriptors.

### 3.3 Captioning and Text Embedding

After mapping a scene, CapNav converts confirmed 3D tracks into language-aware object entries. For each track $T_j$, we collect associated 2D crops $\{I_j^{(k)}\}$ and the Detic-predicted class label. A simple multi-view heuristic (e.g., frontal, high-resolution, minimally occluded views) selects one or a few representative crops for captioning.

**Caption prompt and embedding.** Selected crops are sent to a Gemini 2.5 Flash Lite vision–language model with a fixed instruction that enforces a schema:

> "You will receive up to five full-frame indoor RGB images of the SAME tracked object. Each image highlights the object with a visible bounding box describe ONLY that object. The provided category hint may be incorrect. Infer the ACTUAL object category; start with the inferred name in UPPERCASE, followed by a comma, and then a single sentence ($\leq 25$ words) describing appearance, color, material, or notable condition. Do not mention uncertainty, other objects, or metadata."

The model outputs captions of the form

$$y_j = \texttt{OBJECT\_NAME, one-sentence visual description.} \tag{13}$$

(e.g., "WHITE SOFA, a white sectional sofa with curved armrests facing a glass coffee table."). We embed each caption $y_j$ using the BGE-M3 sentence encoder Chen et al. (2024) (HuggingFace ID `BAAI/bge-m3`):

$$\mathbf{e}_j = f_{\text{enc}}(y_j), \qquad \tilde{\mathbf{e}}_j = \frac{\mathbf{e}_j}{\|\mathbf{e}_j\|_2}. \tag{14}$$

We store $\tilde{\mathbf{e}}_j$ together with the object's 3D pose, category prediction, and metadata (e.g., instance ID, voxel indices), forming a compact, caption-enriched object database. The same BGE-M3 encoder is used for all navigation instructions at test time.

### 3.4 Language-conditioned Retrieval and Navigation

At test time, the AGENT module maps a natural-language instruction $q$ to a retrieval query over the caption-enriched object database and converts the best match into a navigation goal.

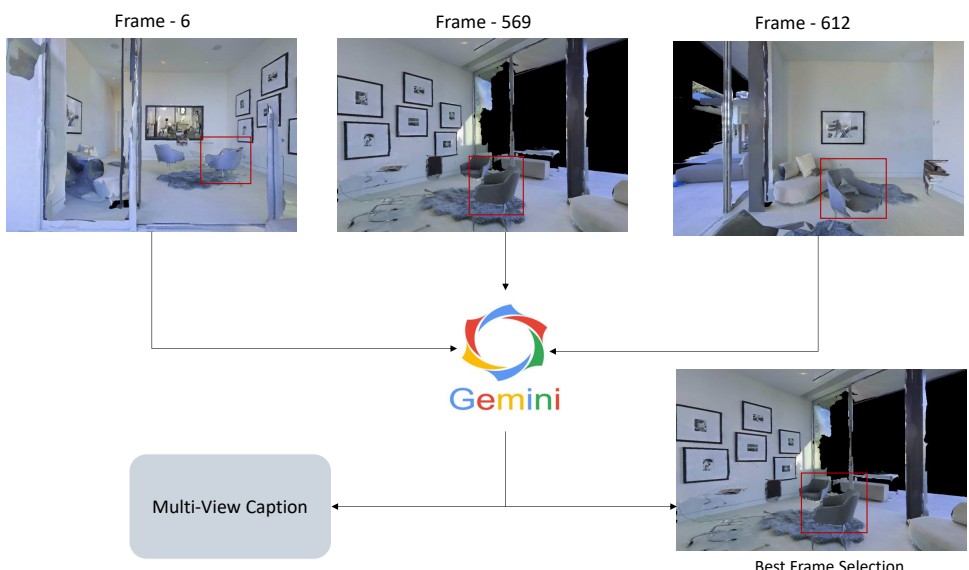

Figure 3: Multi-view selection and captioning. For each confirmed 3D track, CapNav collects image crops across time, selects representative views, and queries a Gemini 2.5 Flash Lite VLM to obtain a concise caption. The caption is then embedded into a sentence-level text space via BGE-M3 and stored with the object's 3D pose.

We first encode $q$ with the same BGE-M3 encoder:

$$\tilde{\mathbf{e}}_q = \frac{f_{\text{enc}}(q)}{\|f_{\text{enc}}(q)\|_2}, \tag{15}$$

and compute semantic similarity for each object $j$:

$$s_j^{\text{sem}} = \tilde{\mathbf{e}}_q^\top \tilde{\mathbf{e}}_j. \tag{16}$$

Objects are ranked by $s_j^{\text{sem}}$; the top-ranked instance is the default target. For ambiguous or safety-critical instructions, we optionally rerank the top-$K$ candidates with Gemini 2.5 Flash Lite, which inspects the corresponding crops and captions and either selects one candidate or rejects the batch.

**Retrieval verification prompt.** For verification, the VLM receives the top-$K$ candidates and is instructed to respond in one of two formats:

> "You are helping a mobile robot find a {object_type}. The objects in this batch are sorted by semantic relevance. For this batch only, your job is: (1) inspect each object's image and caption; (2) decide if any object IS actually a {object_type} (or a very close match); (3) if so, respond *exactly* with 'ACCEPT k: ¡short reason¿' using 1-based indexing; (4) otherwise respond *exactly* with 'REJECT ALL: ¡short reason¿'. Do not pick visually similar but incorrect objects, and do not refer to or compare against any objects outside this batch."

This forces a single accepted instance or a rejection, avoiding hedging across multiple similar objects.

Certain queries use explicit spatial predicates, such as "closest couch" or "farthest table". For these, we first filter objects by type to form candidates $\mathcal{C}$, then measure distances in the 2D grid underlying the voxel map. Let $(r_{\text{R}}, c_{\text{R}})$ be the robot's grid cell and $(r_j, c_j)$ the cell of object $j$:

$$d_j^{\text{grid}} = \sqrt{(r_{\text{R}} - r_j)^2 + (c_{\text{R}} - c_j)^2}, \tag{17}$$

and $d_j^{\text{m}} = s\, d_j^{\text{grid}}$ using map resolution $s$. The nearest valid instance $\hat{j}$ in $\mathcal{C}$ (optionally subject to a semantic threshold) is selected and its grid cell becomes the navigation goal.

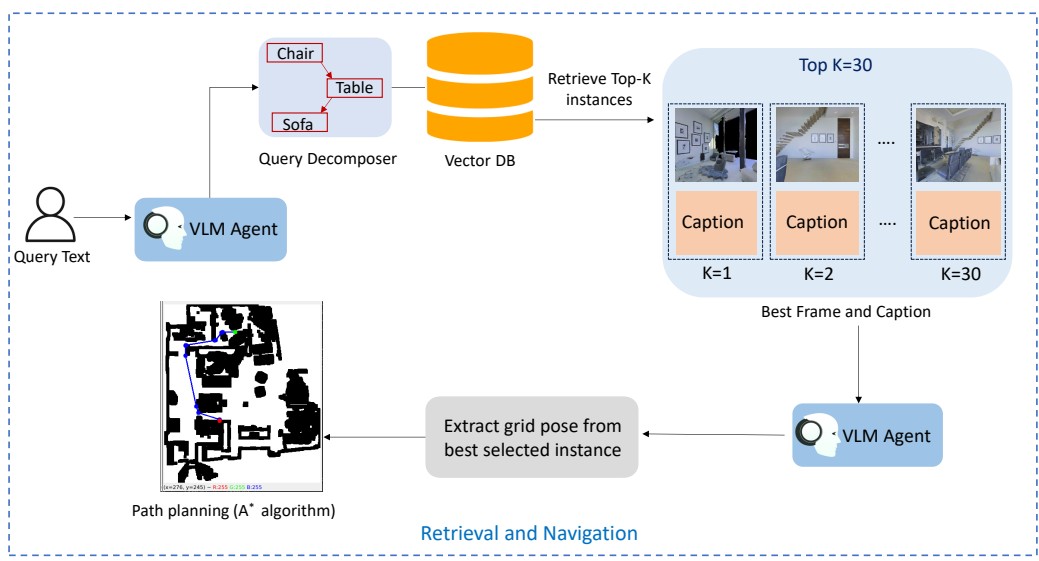

Figure 4: Language-conditioned retrieval and navigation. A free-form instruction is embedded using the same BGE-M3 text encoder as object captions, used to retrieve and optionally Gemini 2.5 Flash Lite–rerank candidate instances, and finally converted into a metric-space goal for planning with A*.

Finally, the selected goal cell is converted into a continuous waypoint, and a standard planner such as A* operates on the voxel grid or a derived occupancy map to compute a collision-free trajectory from the current pose to the target vicinity. Because retrieval operates over caption embeddings that implicitly encode attributes such as color, shape, and material, CapNav can disambiguate visually similar instances and follow fine-grained, description-based instructions without an explicit attribute vocabulary or task-specific training.

## 4 EXPERIMENTS

We evaluate *CapNav* on language-driven multi-object navigation in Habitat Savva et al. (2019) using scenes from Matterport3D Chang et al. (2017). Matterport3D is a large-scale RGB–D dataset with about 10,800 panoramic views (roughly 194,400 RGB–D images) across 90 buildings, providing metrically consistent reconstructions, camera poses, and dense 2D/3D semantics. For each selected scene, we run a virtual exploration pass with a posed RGB–D camera and use ground-truth intrinsics and extrinsics to lift depth into a common world frame. Across ten scenes this produces 11,206 posed RGB–D frames, shared by all methods for map construction. From the semantic annotations we select 31 indoor object categories (e.g., sofa, table, chair, counter) and generate 91 evaluation episodes, each a four-step object-goal sequence, for a total of 364 subgoals. Because VLMaps and related baselines do not support description-based navigation, we use a *label-only, fully automatic evaluation protocol* for fair comparison. All methods receive the same sequence of object *categories* as goals (no free-form descriptions), and a subgoal is successful if the agent base enters a $1\,\mathrm{m}$ ball around any instance of the requested category in the ground-truth map. Metrics are computed automatically from simulator logs and annotations, without human judgement. *CapNav* may internally rely on captions and text embeddings to pick which instance of a category to approach, but evaluation only checks whether the correct semantic label was reached, exactly as for VLMaps.

### 4.1 QUANTITATIVE RESULTS

We compare *CapNav* against three language-conditioned baselines: ZSON Wen et al. (2025), LM-Nav Shah et al. (2023), and CLIP on Wheels (CoW) Gadre et al. (2022) and a strong map-based

Table 1: Multi-object navigation in Habitat. We report success rate (SR, %) for reaching 1–4 sub-goals consecutively and independent subgoal SR. CapNav (ours) achieves the best performance, outperforming VLMaps and all other zero-shot baselines.

| Method | No. subgoals in a row | | | | 2*Indep. |
|---|---|---|---|---|---|
| | 1 | 2 | 3 | 4 | |
| LM-Nav Shah et al. (2023) | 25 | 6 | 0 | 0 | 8.5 |
| ZSON Wen et al. (2025) | 11 | 2 | 0 | 0 | 9.34 |
| CLIP on Wheels (CoW) Gadre et al. (2022) | 22 | 7 | 1 | 0 | 19.8 |
| VLMaps Huang et al. (2023) | 46 | 27 | 16 | 5 | 48.1 |
| **CapNav (ours)** | **62** | **40** | **22** | **14** | **56.6** |

baseline, VLMaps Huang et al. (2023). Following VLMaps, we report two metrics: (i) the *in-a-row success rate* (SR) for reaching 1–4 subgoals consecutively within an episode, and (ii) the *independent subgoal SR*, computed over all 364 subgoals regardless of chain completion. All scores are measured under the same label-only, fully automatic evaluation protocol described in Section 4. Results are summarized in Table 1. ZSON and LM-Nav achieve relatively low success rates across horizons (independent SR $9.34\%$ and $8.5\%$), highlighting the difficulty of long-horizon, object-goal navigation with purely zero-shot or topological language planning. CLIP on Wheels (CoW) improves over these approaches but still trails map-based methods (independent SR $19.8\%$). VLMaps, which reasons directly in a 3D open-vocabulary feature map, is the strongest baseline, with in-a-row SRs of 46, 27, 16, and 5 for one to four subgoals and an independent SR of $48.1\%$. *CapNav* consistently outperforms VLMaps, achieving in-a-row SRs of 62, 40, 22, and 14 and an independent SR of $56.6\%$. This corresponds to gains of $+16$, $+13$, $+6$, and $+9$ points on the four horizons and roughly $+8.5$ points independently. The larger margins on longer chains suggest that caption-enriched, instance-level representations help maintain the correct target across multiple subgoals and reduce compounding errors in cluttered, attribute-rich scenes.

## 4.2 QUALITATIVE RESULTS

We also analyze how *CapNav* uses attribute-aware grounding in cluttered environments. Figure 5 shows representative instructions together with the top-down map, executed trajectory, retrieved viewpoint, and Gemini 2.5 Flash Lite's textual rationale. For queries such as "go to the white sofa with curved edges", "go to the brick wall behind the TV", or "go to the biggest sofa in the map", *CapNav* correctly disambiguates between multiple visually similar candidates by leveraging caption-enriched instance embeddings and spatial clustering, then plans a direct path to the selected goal. The VLM explanations reference the same cues (color, material, shape, size, and local context) that drive internal retrieval, making the system's decisions more interpretable than category-only baselines like VLMaps, which operate purely on dense feature scores without exposing instance-level rationales.

## 5 ABLATION STUDY

We ablate two components in *CapNav*: (i) the *AGENT* vision–language reranking module, which uses a Gemini model to verify retrieved candidates, and (ii) the *attribute heuristic* that decides when to interpret an instruction as a "closest" query versus a pure semantic query. All ablations are run on the same 4-object instruction chains as in Section 4. We report (i) *Success Rate*, which requires completing the full 4-object chain, and (ii) *Subgoal Success Rate*, computed per object in the sequence.

As shown in Table 2, removing AGENT reranking roughly halves the full-chain Success Rate ($15.4\% \rightarrow 7.7\%$) while only mildly reducing the per-subgoal Success Rate ($56.6\% \rightarrow 51.9\%$). This confirms that AGENT effectively filters false positives from retrieval before the planner commits. Disabling the attribute heuristic yields results close to the full model ($15.4\% \rightarrow 15.1\%$), indicating that CapNav is not sensitive to heuristic tuning and that semantic retrieval remains effective even without the heuristic.

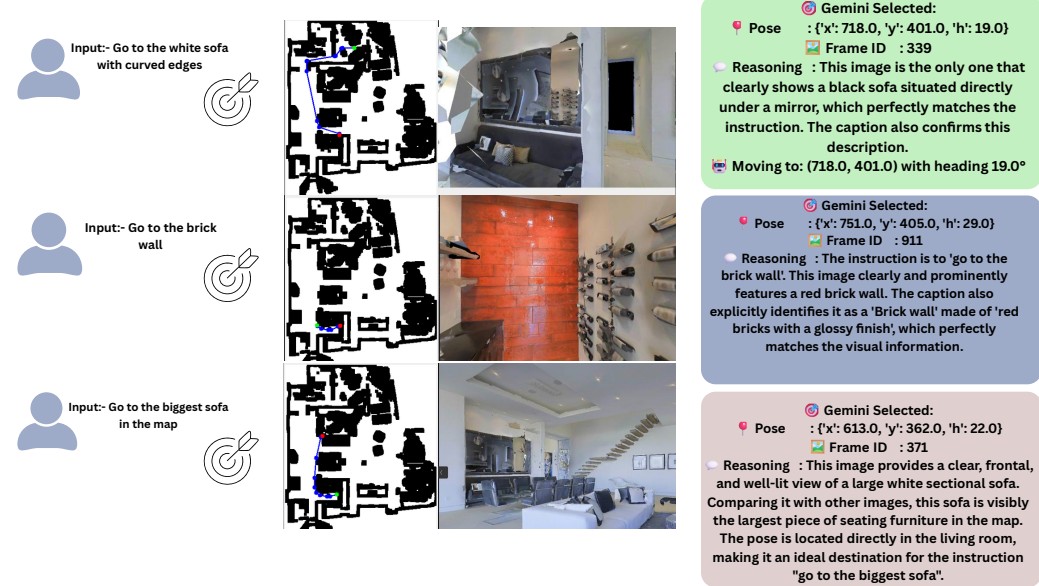

Figure 5: Qualitative result of natural language–based navigation with *CapNav*. Each row shows the user instruction (left), the top-down map and executed trajectory (middle), and the retrieved viewpoint plus Gemini's textual reasoning (right).

Table 2: Ablation of components in CapNav on multi-object navigation. We report full-chain Success Rate and per-subgoal Success Rate.

| Method | Success Rate ↑ | Subgoal Success Rate ↑ |
|---|---|---|
| CapNav (full) | 15.4% | 56.6% |
| w/o AGENT reranking | 7.7% | 51.9% |
| w/o attribute heuristic | 15.1% | 56.0% |

## 6 CONCLUSION AND FUTURE WORK

CapNav is a description-first, instance-centric navigation framework that ties natural-language instructions directly to persistent 3D object instances. From RGB–D streams with known poses, the system builds a semantic voxel map with LSeg features, maintains 3D tracks using a semantic–geometric tracker on Detic detections, and turns each stabilized track into a caption-enriched entry via Gemini 2.5 Flash Lite and BGE-M3. At test time, free-form instructions are embedded in the same text space, used to retrieve a target instance, optionally verified by an AGENT reranking module, and converted into a metric-space waypoint followed by A* planning. Under a label-only, fully automatic evaluation protocol in Habitat with Matterport3D scenes, CapNav outperforms category-only and map-based baselines (ZSON, LM-Nav, CLIP on Wheels (CoW), VLMaps) on multi-object navigation, especially on longer instruction chains in cluttered, attribute-rich environments. Ablations highlight the role of AGENT reranking in stabilizing long-horizon behavior and the benefits of caption-enriched, instance-level representations for interpretable retrieval. Future work will focus on extending CapNav as a deployable embodied system capable of operating autonomously in practical indoor settings. This includes supporting more conversational and context-rich instructions, scaling to larger and dynamic spaces with persistent instance awareness, and executing on physical mobile robots equipped with real-time SLAM, active viewpoint selection, and light-weight interaction capabilities. These extensions aim to translate CapNav's instance-centric, language-grounded navigation from controlled simulation to robust real-world performance without altering the core principles that make the framework effective.

## REPRODUCIBILITY STATEMENT

We take several concrete steps to make our work easy to reproduce and extend.

**Datasets and Simulator.** All experiments are conducted in Habitat Savva et al. (2019) using publicly available Matterport3D Chang et al. (2017) scenes. We specify in Appendix A the exact list of scenes, the virtual exploration policy used to collect RGB–D streams, and the instruction templates for the 91 four-step object-goal episodes (364 subgoals). The paper reports the total number of posed RGB–D frames (11,206) and the 31 semantic categories used for evaluation, so the evaluation set can be reconstructed from Matterport3D annotations.

**Code and Artifacts.** Upon acceptance, we will release the full CapNav codebase, including: (i) map-building scripts (semantic lifting with LSeg + SG3DTracker instance construction), (ii) multi-view crop selection and caption generation with Gemini 2.5 Flash Lite, (iii) BGE-M3-based retrieval and the AGENT reranking module, and (iv) evaluation scripts for generating episodes, running agents, and computing all metrics. We will also provide precomputed voxel maps, 3D tracks, captions, and text embeddings for all evaluation scenes, enabling reproduction of results without rerunning detection or captioning.

**Models and Hyperparameters.** We clearly document all external models used in CapNav: Detic for open-vocabulary detection, LSeg for per-pixel semantics, BGE-M3 (`BAAI/bge-m3`) for text embeddings, and Gemini 2.5 Flash Lite (`gemini-2.5-flash-lite`) for captioning and optional reranking. Tracking thresholds, association weights, clustering radii, retrieval top-$k$, MMR parameters, and A* planner settings are summarized in the main text (Section 3) and detailed in Appendix C, with configuration files in the code repository.

**Evaluation Protocol.** To keep comparisons with VLMaps and other baselines fair, we use a label-only, fully automatic evaluation protocol described in Section 4 and Appendix A: all methods receive the same category-only goal sequences, success is defined by a fixed $1\,\mathrm{m}$ reach threshold around annotated goal instances, and metrics (in-a-row Success Rate and independent Subgoal Success Rate) are computed from simulator logs without human judgements. We release the episode definitions, instruction strings, and metric scripts so that all reported numbers in Tables 1 and 2 can be recomputed exactly.

**Baselines.** ZSON, LM-Nav, CLIP on Wheels (CoW), and VLMaps are either adapted from public code or reimplemented within a shared Habitat harness. For each baseline, we document the checkpoint, configuration, and any deviations from the original papers. All baselines use the same scenes, RGB–D streams, and episodes as CapNav.

**Runtime and Resources.** All experiments are run on a single NVIDIA RTX 3090 GPU with a 32-core CPU. In our setup, scene-level map construction (semantic lifting + SG3DTracker + captioning) takes approximately 8–10 minutes per Matterport3D scene, while retrieval and planning for a single instruction complete in under one second. Exact software versions (CUDA, PyTorch, Habitat, Detectron2/Detic, and tokenizer libraries) are listed in Appendix C and encoded in the released environment files.

**Stochasticity and External Services.** We fix random seeds for Python, NumPy, PyTorch, and Habitat in all experiments and disable nondeterministic CUDA operations where possible. The core mapping, retrieval, and planning pipeline is otherwise deterministic. Since Gemini 2.5 Flash Lite is an external service, its outputs may vary over time; to mitigate this, we release the cached captions and AGENT reranking decisions used in all reported experiments so that our quantitative results can be reproduced exactly even if the underlying API changes.

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
