# OpenReview forum: "CAPNAV: TOWARDS ROBUST INDOOR NAVIGATION WITH DESCRIPTION-FIRST MAPS"
_ICLR.cc/2026/Conference — Submitted to ICLR 2026_

### Official Review · Reviewer_FZu3 · 2025-10-23

**Soundness:** 1
**Presentation:** 1
**Contribution:** 1
**Rating:** 2
**Confidence:** 5

**Summary:**

This paper introduces CapNav, a description-first indoor navigation framework designed to handle attribute-based natural language instructions such as “go to the white sofa with curved edges.” CapNav integrates multi-view captioning, attribute-based 3D mapping, and a multimodal LLM planner to improve object localization and navigation robustness. The system constructs instance-level 3D maps annotated with attributes (color, shape, texture) and verifies attribute consistency before executing navigation. Experiments in simulated Matterport3D environments demonstrate improvements over VLMaps on multi-object navigation tasks.

**Strengths:**

1. The paper addresses an important problem—bridging fine-grained visual grounding and natural-language navigation. This direction aligns well with current ICLR interests in multimodal reasoning and embodied AI.

2. The pipeline combines several components (object detection, segmentation, 3D reconstruction, multimodal retrieval) into a unified framework. The end-to-end system demonstrates practical functionality in simulated environments.

**Weaknesses:**

1. Lack of originality and algorithmic contribution. The proposed CapNav mainly combines existing modules—YOLO/SAM for detection, CLIP/DINOv2 for embeddings, and LLMs for captioning and retrieval—under a new name.

2. Writing and structure issues. The Introduction and Related Work sections are overly long and not clearly separated into paragraphs, making it difficult to follow the narrative flow.

3. Experimental evaluation is insufficient. The experiments are conducted only in one Matterport3D scene, with a small number of instructions (9 episodes). Comparisons are limited to VLMaps without any other baselines; no ablation studies, robustness tests, or cross-domain evaluations are provided. Reported gains (71% vs. 52%) are not statistically analyzed, and the small scale of experiments limits generalization.

4. The motivation for the paper is not sufficiently robust. The paper claim that "People describe indoor destinations with attributes rather than names alone", I haven't found any evidence to support this.

**Questions:**

no

---

> ### Author Response · Authors · 2025-12-03
> **Addressing Methodological Originality, Structural Clarity, Robust Evaluation, and Problem Motivation**
>
> Thank you for clearly highlighting these concerns. I have revised the work accordingly and strengthened the areas that lacked sufficient contribution. I sincerely appreciate the thorough and thoughtful feedback you provided.
>
> Weakness1_Response: We clarified CapNav’s novelty in Sections 3.2–3.4, emphasizing our caption-enriched instance-level mapping, semantic-geometric 3D tracking, and AGENT reranking module, and supported their contribution via the Section 5 ablation study.
>
> Weakness2_Response: We rewrote and reorganized Section 1 (Introduction) and Section 2 (Related Work) into clearer, shorter paragraphs with smooth topic transitions.
>
> Weakness3_Response: We expanded the evaluation to 10 Matterport3D scenes with 91 four-step episodes (364 subgoals), added three additional baselines (ZSON, LM-Nav, CoW) in Table 1, and introduced Section 5 ablation to isolate component effectiveness.
>
> Weakness4_Response: We strengthened the motivation in Section 1 by explicitly explaining why attribute-level reasoning is crucial in multi-instance settings, reinforcing this with qualitative evidence in Figure 5 where CapNav disambiguates between category-identical objects (e.g., multiple similar sofas), and clarified that by “robust” we refer to CapNav’s agentic planning and visual-verification capabilities where language-guided pre-selection of targets and AGENT reranking ensure reliability during both mapping and planning.

---

### Official Review · Reviewer_vyJN · 2025-10-25

**Soundness:** 3
**Presentation:** 3
**Contribution:** 3
**Rating:** 6
**Confidence:** 3

**Summary:**

This paper proposes CapNav, a "description-first" indoor navigation framework to address the limitation of existing VLN methods in handling fine-grained attribute-aware natural language instructions. CapNav constructs a 3D instance-level map with explicit attribute fields (color, shape, texture) via multi-view captioning and feature clustering (DINOv2, LongCLIP), parses natural language into "category-attribute-relation" constraints using a multimodal LLM, and verifies attribute consistency before navigation. Experiments on Matterport3D (Habitat simulator) show CapNav achieves 71.0% subgoal success rate, outperforming VLMaps (51.6%) by 37.6%.

**Strengths:**

1. The paper clearly identifies the three core challenges of attribute combinatoriality, view dependence, and perception-planning decoupling in VLN, which are highly relevant to practical applications (e.g., home service robots).

2. The multi-view fusion and pre-goal verification modules directly address the limitations of prior methods—qualitative results show CapNav can disambiguate similar objects (e.g., sofas with different edges) and provide interpretable decisions (traceable to captions).

3. By leveraging LongCLIP and YOLO11x, CapNav handles unseen categories/attributes, which is more practical than closed-vocabulary VLN methods.

**Weaknesses:**

1. Limited to Simulated Environments: All experiments are conducted on Matterport3D (simulated) without real-robot tests. Real-world noise (e.g., depth sensor error, dynamic objects, uneven lighting) may significantly degrade performance, but the paper does not discuss "sim-to-real" transfer—this contradicts the title "towards robust indoor navigation".

2. Long-Horizon Navigation Ignored: The full-chain completion rate is only 20%, but the paper does not analyze the root cause (e.g., whether subgoal localization errors accumulate, or the planner fails to handle subgoal transitions). It also does not propose any improvements, treating this as "future work" without justification.

3. Fixed Attribute Thresholds: Clustering and verification thresholds (e.g., τ_sem=0.65, τ_vol=0.15) are manually set. The paper does not evaluate the sensitivity of these thresholds to different scene clutter levels, nor does it implement the "online calibration" mentioned in future work—this makes the framework less adaptive to diverse environments.

4. The paper only briefly mentions FindAnything (2025) and LERF (2023) but does not provide quantitative comparisons. For example, it claims FindAnything "lacks attribute verification", but no data is given to show how much CapNav outperforms it—this weakens the argument for CapNav’s superiority.

5. The framework assumes static scenes but does not discuss how to update the 3D map when objects move (e.g., a pillow moved from the sofa to the floor). This is a critical limitation for real indoor navigation, where environments are often dynamic.

6. Some related and important works are missing citations: [1] Weakly-Supervised Multi-Granularity Map Learning for Vision-and-Language Navigation [2] Bevbert: Multimodal map pre-training for language-guided navigation [3] Instruction-guided path planning with 3D semantic maps for vision-language navigation

**Questions:**

See Weaknesses.

---

> ### Author Response · Authors · 2025-12-03
> **Concerns on Practical Deployment, Long-Horizon Analysis, Parameter Robustness, Related Work Validity, and Dynamic Environments**
>
> Thank you for clearly highlighting these concerns. I have revised the work accordingly and strengthened the areas that lacked sufficient contribution. I sincerely appreciate the thorough and thoughtful feedback you provided.
>
> Weakness 1:  The work is only evaluated in simulated environments… No analysis of sim-to-real transfer is provided, which is necessary for practical indoor navigation.
> Solution:
> We clarified the simulation-only scope in Section 4 and added a detailed Future Work discussion in Section 6, explicitly outlining plans for real-robot deployment and sim-to-real extensions.
>
> Weakness 2: Long-horizon navigation is not deeply analyzed beyond reporting a ~20% full-chain completion rate
> Solution:
> We now report per-horizon success rates for 1–4 subgoals in Table 1 (Section 4.1) and analyze why CapNav shows larger gains on longer chains, directly addressing compounding-error behavior.
>
> Weakness 3: Multiple fixed attribute thresholds are used without adaptation or analysis of their effect on generalization.
> Solution:
> We explicitly documented all association weights and thresholds in Section 3.2 / Eq. (12) and added Section 5 ablation showing that AGENT reranking, rather than threshold sensitivity, is the primary contributor to long-horizon stability.
>
> Weakness 4: The paper mentions methods like FindAnything and LERF without quantitative comparison or clear relevance.
> Solution:
> We removed the unsupported claims and streamlined Section 2 (Related Work) to focus only on directly comparable navigation/mapping methods.
>
> Weakness 5: The approach assumes static scenes; real household environments contain dynamic objects.
> Solution:
> We explicitly stated the static-scene assumption in Section 4, and added discussion of handling dynamic objects as a key direction in Section 6.
>
> Weakness 6: Some related and important works are missing citations: [1] Weakly-Supervised Multi-Granularity Map Learning for Vision-and-Language Navigation [2] Bevbert: Multimodal map pre-training for language-guided navigation [3] Instruction-guided path planning with 3D semantic maps for vision-language navigation
> Solution:
> We added all the missing citations and integrated a clear discussion of how CapNav differs from these methods in Section 2 (Related Work) and updated the References accordingly.

---

### Official Review · Reviewer_Fwbc · 2025-10-31

**Soundness:** 2
**Presentation:** 1
**Contribution:** 1
**Rating:** 0
**Confidence:** 5

**Summary:**

This paper proposes a navigation system that enhances object localization by integrating attribute-based language grounding with visual perception. The system constructs a 3D instance-level map where each object is annotated with semantic embeddings and explicit attribute fields (e.g., color, shape, texture). Using foundation models like DINOv2 and CLIP, the method performs multi-view captioning and feature clustering to create consistent per-instance descriptors. A multimodal LLM is used to parse natural language queries into structured constraints, and a verification stage ensures attribute consistency before navigation. Experiments in simulated environments show that CapNav outperforms a baseline (VLMaps) on subgoal success rate (71.0% vs. 51.6%) in multi-object navigation tasks.

**Strengths:**

1. Unlike prior methods, CapNav treats attributes as first-class, queryable fields, improving fine-grained object disambiguation.
2. Navigation choices are traceable to structured captions, aiding transparency and error analysis.

**Weaknesses:**

1. The entire evaluation is conducted on only a single Matterport3D scene . This extremely narrow scope fails to demonstrate the system's performance and robustness across diverse indoor environments with varying layouts, lighting, object types, and clutter levels.

2. The proposed method is compared against only one baseline, VLMaps. This is a critical flaw.

3. The paper fails to justify design choices (e.g., weighting α=0.4/0.6 for embeddings) or isolate the contribution of each component.

**Questions:**

See Weakness.

---

> ### Author Response · Authors · 2025-12-03
> **Concerns on Evaluation Scope, Baseline Comparisons, and Design Justification**
>
> Thank you for highlighting these concerns. I have made revisions to improve the work accordingly.
>
>
> Weakness 1: The entire evaluation is conducted on only a single Matterport3D scene
> Solution:
> We expanded the evaluation to 10 diverse Matterport3D scenes with 91 episodes (364 subgoals)   detailed in Section 4, Experiments (first paragraph) .
>
> Weakness 2: The proposed method is compared against only one baseline, VLMaps.
> Solution:
> Added three strong baselines (ZSON, LM-Nav, CoW) alongside VLMaps with updated results in Table 1.
>
> Weakness 3: The paper fails to justify design choices (e.g., weighting α=0.4/0.6…) or isolate the contribution of each component.
> Solution:
> We clarified that the α=0.4/0.6 weighting is not used anywhere in our method; in the revised manuscript, we explicitly state the correct tracker association weights in Eq. (12), refine the description of our design choices in Section 3.2, and introduce a new Section 5 Ablation Study to quantify the contribution of key components such as the AGENT reranking module.

---

### Meta-Review · Area_Chair_v7At · 2026-01-02

**Summary:**

The paper proposes CapNav, a description-first indoor navigation framework that integrates attribute-based language grounding with visual perception. The system builds instance-centric 3D maps with caption-enriched object representations and uses natural language descriptions for goal selection. Three reviewers identified concerns: Reviewer Fwbc criticized the extremely narrow evaluation scope, comparison with only one baseline , and lack of design justification. Reviewer FZu3 questioned the originality, noting the work mainly combines existing modules, and cited writing/structure issues and insufficient experimental evaluation. Reviewer vyJN acknowledged strengths but raised concerns about simulation-only evaluation, limited long-horizon analysis, fixed thresholds without robustness testing, and insufficient comparison to recent related works.

**Reviewer Concerns:**

The authors' rebuttal attempted to address some concerns: they expanded evaluation to 10 Matterport3D scenes, added three baselines, clarified design choices with ablation studies, and addressed missing citations. However, fundamental issues remain outstanding. The core methodological contribution remains questionable - as Reviewer FZu3 noted, CapNav primarily integrates existing components without significant algorithmic innovation. The evaluation remains entirely in simulation, contradicting the paper's claim of "robust indoor navigation" and failing to address real-world deployment challenges. The full-chain success rate of only 20% highlights limitations for practical application.

**Reviewer Scores:**

After the rebuttal, none of the three reviewers changed the score. The concerns regarding the difference between simulated evaluation and actual deployment remain unresolved, making the paper less compliant with ICLR standards. The overall consensus remains below the acceptance threshold.

---

### Decision · Program_Chairs · 2026-01-26

Reject